# Spontaneous Attenuation of Alcoholic Fermentation via the Dysfunction of Cyc8p in *Saccharomyces cerevisiae*

**DOI:** 10.3390/ijms25010304

**Published:** 2023-12-25

**Authors:** Daisuke Watanabe, Maika Kumano, Yukiko Sugimoto, Hiroshi Takagi

**Affiliations:** 1Division of Biological Science, Graduate School of Science and Technology, Nara Institute of Science and Technology, 8916-5 Takayamacho, Ikoma 630-0192, Nara, Japanhiro@bs.naist.jp (H.T.); 2Institute for Research Initiatives, Nara Institute of Science and Technology, 8916-5 Takayamacho, Ikoma 630-0192, Nara, Japan

**Keywords:** alcoholic fermentation, Cyc8p–Tup1p complex, [*GAR^+^*], glucose repression, non-genetic heterogeneity, pyruvate decarboxylase, *Saccharomyces cerevisiae*

## Abstract

A cell population characterized by the release of glucose repression and known as [*GAR^+^*] emerges spontaneously in the yeast *Saccharomyces cerevisiae*. This study revealed that the [*GAR^+^*] variants exhibit retarded alcoholic fermentation when glucose is the sole carbon source. To identify the key to the altered glucose response, the gene expression profile of [*GAR^+^*] cells was examined. Based on RNA-seq data, the [*GAR^+^*] status was linked to impaired function of the Cyc8p–Tup1p complex. Loss of Cyc8p led to a decrease in the initial rate of alcoholic fermentation under glucose-rich conditions via the inactivation of pyruvate decarboxylase, an enzyme unique to alcoholic fermentation. These results suggest that Cyc8p can become inactive to attenuate alcoholic fermentation. These findings may contribute to the elucidation of the mechanism of non-genetic heterogeneity in yeast alcoholic fermentation.

## 1. Introduction

In multicellular organisms, a single-origin cell proliferates and differentiates into a variety of cell types. Even unicellular organisms, such as the model eukaryote *Saccharomyces cerevisiae*, genetically or epigenetically generate phenotypic heterogeneity in growth rate, stress response, gene expression, and cellular metabolism within a population [1,2,3,4,5]. Studies of metabolic diversity in *S. cerevisiae* have achieved significant impacts over the years because of their broad implications, ranging from tumor biology to applied microbiology [6,7,8]. Although diversity confers many advantages to individual cells in natural environments, the metabolic heterogeneity of *S. cerevisiae* is unpredictable and undesirable for industrial fermentation processes [9,10]. 

[*GAR^+^*] is a non-genetic element that causes metabolic heterogeneity in *S. cerevisiae* and other yeast species [11,12,13,14,15,16]. Although yeast cells can assimilate various carbon sources, glucose represses the transcription of genes involved in the utilization of alternative carbon sources. This phenomenon, termed glucose repression, makes yeast preferentially use glucose, which is the most efficient energy source as a glycolytic substrate [17,18]. Therefore, *S. cerevisiae* cannot form colonies on glycerol-medium plates that contain glucosamine, a nonmetabolizable glucose mimetic. However, spontaneous colonies that can use glycerol in the presence of glucosamine emerge at low frequencies (e.g., 10^−1^ to 10^−4^). This phenotype that is due to defects in glucose repression is inherited in a non-Mendelian manner via the prion-like [*GAR^+^*] factor [11]. Regarding the physiological significance of the [*GAR^+^*] phenotype, this is considered an adaptive strategy in response to environmental cues through conversion from a carbon metabolic specialist to a generalist [12]. Retarded alcoholic fermentation is another well-studied phenotype in [*GAR^+^*] isolates and is possibly associated with less effective glucose utilization [13,14]. Controlling the emergence of [*GAR^+^*] variants may contribute to a more stable alcoholic fermentation at the industrial scale, although the molecular entity and formation process of [*GAR^+^*] are not fully understood.

In glucose repression in *S. cerevisiae*, the Cys2His2 zinc-finger transcriptional repressor Mig1p has a pivotal role [17,18]. In the presence of glucose, Mig1p is dephosphorylated by the Reg1p–Glc7p protein phosphatase complex and is localized in the nucleus. Nuclear Mig1p binds to the promoters of genes involved in the utilization of alternative carbon sources, such as sucrose (*SUC2*), maltose (*MAL* genes), and galactose (*GAL* genes), where it recruits the Cyc8p–Tup1p corepressor complex [19]. Cyc8p–Tup1p, a highly conserved complex that consists of one Cyc8p and four Tup1p subunits, mediates the repression of target genes by interacting with hypoacetylated histones, histone deacetylases, and the RNA transcriptional machinery [20,21]. Cyc8p–Tup1p cooperates not only with Mig1p but also with a variety of transcriptional regulators involved in mating, meiosis, sporulation, flocculation, DNA repair, osmotic response, and oxygen response [20,22,23]. This complicated regulatory pathway of glucose repression should be carefully explored to determine the molecular switch of [*GAR^+^*].

In this study, we identified the factor that is closely related with the [*GAR^+^*] phenotype, based on the profiles of alcoholic fermentation and gene expression. Moreover, how [*GAR^+^*] affects alcoholic fermentation was examined using metabolome analysis. Our findings provide important clues to understand non-genetic heterogeneity during alcoholic fermentation.

## 2. Results

### 2.1. Spontaneous [GAR^+^] Retards Alcoholic Fermentation from Glucose

To examine whether the reported [*GAR^+^*] phenotype of delayed alcoholic fermentation [13,14] is associated with cellular responses to glucose, we performed fermentation tests of a *S. cerevisiae* sake strain K701 ([*gar^−^*] strain) and its spontaneous [*GAR^+^*] isolate [14] under different carbon conditions (Figure 1). It should be noted that the [*GAR^+^*] phenotype of the used strain is unlikely caused by a chromosomal mutation because we found that this strain lost the phenotype during the long-term culture under fermentative conditions (e.g., 5 d cultivation in yeast extract–peptone (YP) medium containing 20% glucose). The [*GAR^+^*] strain was precultured in a glycerol/glucosamine-containing medium to prevent the phenotypic reversion. While [*GAR^+^*] led to a marked decrease in the fermentation rate when glucose was used as the sole carbon source, the delay was alleviated in mixed carbon sources of glucose and sucrose. Release from glucose repression by [*GAR^+^*] may allow the use of sucrose, compensating for the defect in glucose consumption. Similar data were obtained from the laboratory strain of the X2180 background (Appendix A). If the physiological significance of [*GAR^+^*] only enhances adaptation to mixed carbon environments, interpreting the observed difference in the pure glucose medium remains challenging. We should shed more light on the effect of [*GAR^+^*] on cellular responses to glucose.

### 2.2. [GAR^+^] Alters Glucose Response via the Cyc8p–Tup1p Complex

To identify the key to the altered glucose response in the [*GAR^+^*] strain, we performed RNA-seq analysis. In a previous report [11,15], the gene expression profile of the [*GAR^+^*] strain was examined using glucose-grown cultures immediately before the diauxic shift. Consequently, the hexose transporter gene *HXT3* was identified as the only differentially expressed gene between the [*gar^−^*] and [*GAR^+^*] strains. Since glucose is rapidly consumed during the log phase and is almost completely depleted at the diauxic shift, the experimental condition used may be suboptimal for the analysis of glucose responses. In the present study, we focused on the transcriptomic changes in glycerol-grown cultures with the addition of glucosamine, a nonmetabolizable glucose analog in *S. cerevisiae* (Appendix A). 

We began assessing the transcriptomic data with the representative genes for the use of non-glucose carbon sources subjected to glucose repression (Figure 2A). Addition of glucosamine severely repressed the expression of *SUC2* (encoding invertase, a sucrose-hydrolyzing enzyme), *MAL32* (maltase for maltose catabolism), and *ADH2* (alcohol dehydrogenase for the conversion of ethanol to acetaldehyde) to 5–12% in the [*gar^−^*] strain of the X2180 background. In the [*GAR^+^*] strain, the transcriptional repression by glucosamine was partially canceled and the mRNA levels increased 2–7 times by [*GAR^+^*] in the presence of glucosamine. These data confirmed the glucosamine-triggered gene repression in the [*gar^−^*] strain and its release in the [*GAR^+^*] strain.

Next, we investigated the genes with more sharply downregulated (203 genes; Appendix A) or upregulated (183 genes; Appendix A) expression by glucosamine in the [*gar^−^*] strain than in the [*GAR^+^*] strain. These are possible candidate genes whose glucose responses are attenuated by [*GAR^+^*]. Most of the genes with glucosamine-downregulated expression were involved in the cell cycle, cell wall, and ribosome, which are essential for cell growth (Appendix A). The different responses between the [*gar^−^*] and [*GAR^+^*] strains are likely due to glucosamine-triggered growth inhibition specifically in the [*gar^−^*] strain [11]. Among the genes with glucosamine-upregulated expression, the hexose transporter-encoding *HXT4* and *HXT3* were markedly derepressed in the [*GAR^+^*] strain with or without glucosamine (Appendix A and Figure 2B). The glucose-responsive transcription factor Rgt1p, together with the Cyc8p–Tup1p complex, represses the expression of hexose transporter genes in the absence of glucose [22]. Thus, [*GAR^+^*] may abolish the Rgt1p-dependent transcriptional repression. Osmotic stress-responsive genes targeted by the Hog1p MAPK pathway, such as *GRE1*, *SIP18*, and *PAI3*, also showed sharply upregulated expression in the [*gar^−^*] strain, although their expression levels were low in the [*GAR^+^*] strain, regardless of glucosamine addition (Appendix A and Figure 2C). Upon phosphorylation by Hog1p in response to glucose, the transcription factor Sko1p induces gene expression in cooperation with Cyc8p–Tup1p [23]. Thus, [*GAR^+^*] may abolish Sko1p-dependent transcriptional induction.

In summary, [*GAR^+^*] caused pleiotropic changes in the transcriptomic responses to glucosamine: defects in (i) repression of glucose-repressible genes, (ii) Rgt1p-dependent repression of hexose transporter genes, and (iii) Sko1p-dependent induction of osmotic stress-responsive genes. We realized that all these transcriptional controls involve the Cyc8p–Tup1p complex. [*GAR^+^*] may be closely associated with the dysfunction of this transcriptional corepressor/coactivator. Notably, the *CYC8* gene was also listed as a differentially expressed gene (Appendix A); the expression of *CYC8* in the presence of glucosamine exhibited clear repression in the [*gar^−^*] strain and partial derepression in the [*GAR^+^*] strain. This might be associated with the fact that *CYC8* is one of the genes directly targeted by Sko1p [24].

### 2.3. Dysfunction of Cyc8p Leads to Inactivation of Pyruvate Decarboxylase

To verify whether the [*GAR^+^*] phenotypes originate from defective Cyc8p–Tup1p functions, we performed fermentation tests of the *CYC8*- or *TUP1*-deleted strain in a laboratory strain of the BY4741 background under different carbon conditions. While *cyc8*Δ led to a marked decrease in the fermentation rate in the glucose medium, the delay was alleviated in the mixed carbon sources of glucose and sucrose (Figure 3). This trend was fully consistent with that observed for the [*GAR^+^*] strain. Deletion of *TUP1* caused a severe delay in alcoholic fermentation under both carbon conditions (Appendix A).

Since deletion of the *CYC8* gene severely affects cell growth under standard laboratory growth conditions [25,26,27], retarded alcoholic fermentation may result from growth delay in the glucose medium. In fact, however, *cyc8*Δ grew to a level comparable to that of the wild type under the fermentation conditions used at 27 h from the onset of the fermentation test. The mutant strain was grown more highly than the wild-type strain at the end of the 5-day fermentation test. We should be aware that the growth phenotypes in fermentation environments sometimes exhibit unexpected behavior compared to those in laboratory conditions.

To reveal whether Cyc8p affects the fermentation rate via the altered control of carbon metabolism, we analyzed the intracellular metabolite profile of *cyc8*Δ cells at 27 h from the onset of the fermentation test (Figure 4A). While the levels of most glycolytic intermediates were largely unaffected by *cyc8*Δ (0.5–1.5 times as high as the wild type), pyruvate was highly accumulated in *cyc8*Δ cells (2.4 times). Although we previously reported that the high fermentation capacity of industrial yeast strains is associated with decreased metabolic flux into the carbon storage pathway [28,29], *cyc8*Δ did not exhibit a predominant change in the levels of glucose 1-phosphate or UDP-glucose. This suggests that Cyc8p controls alcoholic fermentation via an unknown mechanism. Notably, among redox couples, NADH showed the highest level in *cyc8*Δ cells (3.3 times). Based on these data, we speculate that Cyc8p targets pyruvate decarboxylase, which degrades pyruvate into acetaldehyde and carbon dioxide (Figure 4B). Inactivation of pyruvate decarboxylase may also account for the decreased emission of carbon dioxide. Since *S. cerevisiae* produces ethanol from acetaldehyde to regenerate NAD^+^ for glycolysis, the impaired activity of pyruvate decarboxylase may lead to decreases in acetaldehyde production and the NAD^+^/NADH ratio. Neither pyruvate nor NADH was highly accumulated in *tup1*Δ cells (Appendix A), suggesting distinct roles for Cyc8p and Tup1p in the glycolytic control.

## 3. Discussion

The stochastic decrease in the alcoholic fermentation capacity of *S. cerevisiae* presents an industrial challenge because of the unknown and unpredictable nature of this phenomenon. In the present study, we analyzed the transcriptome responses of an *S. cerevisiae* [*GAR^+^*] strain with a delayed fermentation rate that emerges spontaneously even in standard cultures at the laboratory scale [14]. Our RNA-seq data suggest that [*GAR^+^*] is closely associated with loss of function of the Cyc8p–Tup1p complex. The observed [*GAR^+^*] phenotypes in the fermentation tests and metabolome analysis were more similar to *cyc8*Δ than to *tup1*Δ. The independent roles of Cyc8p and Tup1p [24] may be responsible for the phenotypic differences. Together, the spontaneous inactivation of Cyc8p may lead to attenuation of alcoholic fermentation, generating phenotypic heterogeneity to benefit the yeast cell population.

Previous reports have shown that [*GAR^+^*] is transmissible through non-Mendelian genetic elements such as prions [11]. Interestingly, Cyc8p contains two glutamine-rich regions that drive prion formation and is known as a causal protein of the [*OCT^+^*] prion [30]. The [*OCT^+^*] strain exhibits derepression of the glucose-repressed genes and assimilation of non-glucose carbon sources, which were commonly observed in the [*GAR^+^*] strain. The only experimental fact indicating a difference between [*OCT^+^*] and [*GAR^+^*] is that [*OCT^+^*] formation is Hsp104-dependent while [*GAR^+^*] formation is Hsp70-dependent [11]. Brown and Lindquist (2009) [11] proposed that the appearance of [*GAR^+^*] is associated with a protein complex composed of the proton pump Pma1p and the glucose signaling protein Std1p, which acts upstream of Cyc8p–Tup1p. However, our results demonstrated that [*GAR^+^*] affects not only Std1p- and Rgt1p-regulated hexose transporter gene expression but also broader pathways (e.g., glucose repression and the Hog1p-targeted pathway) involving Cyc8p–Tup1p.

Notably, we found a unique deletion mutation in the amino-terminal glutamine-rich region of the *CYC8* gene product in the *S. cerevisiae* Km67 strain, which is characterized as a highly [*GAR^+^*]-inducible strain [14,31] (Appendix A). We confirmed that the deletion mutation in the amino-terminal glutamine-rich region of the *CYC8* gene product in the Km67 strain is unique not only among X2180 and sake strains but also among all known *S. cerevisiae* genomes shown in the *Saccharomyces* Genome Database (SGD; https://www.yeastgenome.org/; accessed on 18 December 2023). In addition, we confirmed that the spontaneous [*GAR^+^*] strains used in this study do not contain any mutations in the *CYC8* gene in comparison to the parental [*gar^−^*] strains, based on Sanger sequencing. To determine the molecule responsible for [*GAR^+^*], prion formation of Cyc8p and/or its closely related factors should be investigated in the future. The spontaneous attenuation of Cyc8p found in the present study will contribute to a more stable and predictable phenotype of the yeast strains used in the alcoholic fermentation industry.

Cellular diversity within a population is essential for adaptation to and survival in changing environments. The superior capacity for alcoholic fermentation allows *S. cerevisiae* to efficiently acquire energy and propagate under glucose-rich conditions. However, yeast cells inactivate various stress tolerance mechanisms to optimize alcoholic fermentation [28,29,32,33], leading to an unprepared status. The spontaneous emergence of [*GAR^+^*] cells with impaired alcoholic fermentation at a low frequency may benefit the whole yeast population in case of drastic environmental changes. The main rising challenge in microbial ecology is to determine how *S. cerevisiae* chooses a small population in which [*GAR^+^*] is expected to be induced.

In the microbial ecosystem, alcoholic fermentation can be interpreted as the selfish behavior of *S. cerevisiae* because it produces ethanol, which kills microbial competitors [34,35,36]. Thus, the induction of [*GAR^+^*] may be a good strategy for microorganisms that aim for symbiosis with *S. cerevisiae*. In fact, coculture of *S. cerevisiae* and some bacterial species leads to a higher induction rate of [*GAR^+^*] [12,13,14]. In the present study, we found evidence supporting that [*GAR^+^*] specifically inactivates pyruvate decarboxylase (Figure 4), an enzyme uniquely present in yeasts and a few other microorganisms capable of alcoholic fermentation [37,38]. Cyc8p–Tup1p may cooperate with Pdc2p, a transcriptional activator that commonly acts on *PDC1* and *PDC5* pyruvate decarboxylase genes [39,40]. Since our transcriptomic data indicated only weak effects of [*GAR^+^*] on *PDC1* and *PDC5* expression, Cyc8p–Tup1p or Cyc8p alone may have posttranslational effects on the glycolytic enzyme activity. Coexisting bacteria may induce [*GAR^+^*] of *S. cerevisiae* to inactivate pyruvate decarboxylase and attenuate alcoholic fermentation for their benefit.

In summary, we revealed the primary role of [*GAR^+^*] as an inhibitor of alcoholic fermentation under glucose-rich conditions. [*GAR^+^*] was defined as the cellular status in which glucose responses are attenuated due to impaired functions of Cyc8p. *S. cerevisiae* cells may spontaneously induce [*GAR^+^*] to generate heterogeneity in the population even from a single cell. Such metabolic heterogeneity contributes to efficient energy production under continuously fluctuating environments. Furthermore, our previous studies have suggested that [*GAR^+^*]-mediated metabolic modification of *S. cerevisiae* is targeted by the other microorganisms in microbial ecosystems. Coexisting bacteria may harness [*GAR^+^*] to regulate toxic ethanol production by *S. cerevisiae* for their adaptation and survival. Based on these perspectives, we conclude that [*GAR^+^*] is a key factor that drives microbial sociality and symbiosis.

## 4. Materials and Methods

### 4.1. Yeast Strains Used in This Study

The prototrophic diploid laboratory yeast strain X2180 was obtained from the American Type Culture Collection (ATCC, Manassas, VA, USA). K701, a representative sake strain, was obtained from the Brewing Society of Japan (Tokyo, Japan). Km67, a strain isolated from kimoto-type sake [31], was kindly provided by Kiku-Masamune Sake Brewing Co., Ltd. (Kobe, Japan). The *S. cerevisiae* laboratory strain BY4741 (*MAT***a** *leu2*Δ*0 his3*Δ*1 ura3*Δ*0 met15*Δ*0*) and its *CYC8* or *TUP1*-deletion mutant (Y07161; *MAT***a** *leu2*Δ*0 his3*Δ*1 ura3*Δ*0 met15*Δ*0 cyc8*Δ*::kanMX* or Y07198; *MAT***a** *leu2*Δ*0 his3*Δ*1 ura3*Δ*0 met15*Δ*0 tup1*Δ*::kanMX*) were purchased from the European *Saccharomyces cerevisiae* Archive for Functional Analysis (Euroscarf, Oberursel, Germany). BY4741 and X2180 are isogenic strains to S288C; BY4741 is part of a set of deletion strains derived from S288C in which commonly used selectable marker genes were deleted [41]. X2180 was generated via diploidization of S288C [42]. Yeast cells were routinely cultured in yeast-extract–peptone–dextrose (YPD) medium (1% yeast extract (Difco Laboratories, Detroit, MI, USA), 2% peptone, and 2% glucose). Spontaneous [*GAR^+^*] strains of X2180 and K701 were isolated in our previous study [14]. The [*GAR^+^*] strains were maintained in yeast-extract–peptone–glycerol (YPGly) medium (1% yeast extract (Difco Laboratories), 2% peptone, and 2% glycerol), supplemented with 0.05% glucosamine for [*GAR^+^*] strains at 30 °C. Growth media reagents were purchased from Nacalai Tesque, Inc. (Kyoto, Japan), unless otherwise stated.

### 4.2. Fermentation Test

For measurements of alcoholic fermentation rates, yeast cells were precultured in YPGly or YPGly supplemented 0.05% glucosamine (for [*GAR^+^*] strains) at 30 °C overnight, inoculated into 50 mL of yeast-extract–peptone (YP; 1% yeast extract (Difco Laboratories), 2% peptone) medium containing 20% glucose or 10% glucose plus 10% sucrose at a final optical density at a wavelength of 600 nm (OD_600_) of 0.1, and further incubated at 30 °C without shaking. The fermentation progression was continuously monitored by measuring the volume of the evolved carbon dioxide gas using a Fermograph II apparatus (Atto, Tokyo, Japan) [43]. Fermentation media reagents were purchased from Nacalai Tesque, Inc., unless otherwise stated.

### 4.3. RNA-Seq Analysis

The X2180 parental [*gar^−^*] strain and its [*GAR^+^*] variant were precultured in YPD medium overnight at 30 °C, inoculated into 80 mL of YPGly medium at a final OD_600_ of 0.1, and further incubated at 30 °C with vigorous shaking for 3 h. Collected log-phase cells were divided into two flasks and cultured in 40 mL of YPGly medium with or without 0.05% glucosamine (Appendix A) at 30 °C with vigorous shaking for 3 h. The cells were then collected, immediately frozen in liquid nitrogen, and subjected to total RNA extraction using an RNeasy Mini Kit (Qiagen, Venlo, The Netherlands) according to the manufacturer’s instructions. RNA-seq analysis was performed at Hokkaido System Science Co., Ltd. (Sapporo, Japan). Briefly, the quantity and quality of total RNA were assessed using a NanoDrop Spectrophotometer (Thermo Fisher Scientific, Waltham, MA, USA) and an Agilent 2100 Bioanalyzer (Agilent Technologies, Santa Clara, CA, USA). After rRNA depletion, a double-stranded cDNA library was prepared and sequenced using an Illumina HiSeq 2000 sequencing system. Approximately 25 million paired-end reads (2 × 100 bp) were sequenced for each sample. The adapter sequences were trimmed using cutadapt (v1.1). The sequencing reads were mapped to the *S. cerevisiae* S288c reference genome using TopHat (v2.0.14)/Bowtie. The expression levels of individual genes were quantified by the number of fragments (paired-end reads) mapped to the coding region of each gene with a value of fragments per kilobase of exon per million fragments mapped (FPKM) using the Cufflinks (v2.2.1) program.

### 4.4. Analysis of Intracellular Metabolite Profiles

During the fermentation tests in YP medium containing 20% glucose, BY4741 wild-type and *cyc8*Δ cells corresponding to an OD_600_ of 20 were collected at 27 h from the onset of the fermentation tests. All pretreatment procedures were performed according to the protocols provided by Human Metabolic Technologies, Inc. (Tsuruoka, Japan). Briefly, each yeast cell sample was washed twice with ice-cold Milli-Q water, suspended in 1.6 mL of methanol containing 5 μM internal standard solution 1 (Human Metabolic Technologies), and sonicated for 30 s at room temperature. Cationic and anionic compounds were measured by capillary electrophoresis–time of flight mass spectrometry (CE-TOFMS). Peaks detected by CE-TOFMS were extracted using the automatic integration software MasterHands (v2.19.0.2; Keio University) [44] to obtain peak information, including *m/z*, migration time, and peak area. The peaks were annotated with putative metabolites from the Human Metabolic Technologies metabolite database based on their migration times in CE and *m/z* values determined by TOFMS. Relative metabolite levels were calculated by normalizing the peak area of each metabolite to the area of the internal standard.

## Figures and Tables

**Figure 1 ijms-25-00304-f001:**
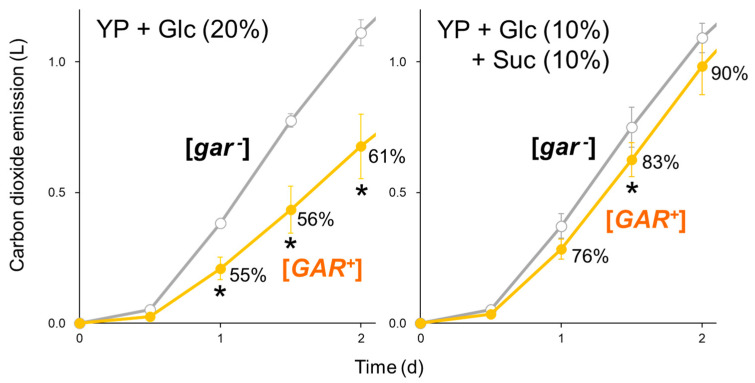
[*GAR^+^*] retards alcoholic fermentation when glucose is the sole carbon source. Carbon dioxide emissions of [*gar^−^*] (gray) and [*GAR^+^*] (orange) strains in the K701 background were monitored in YP + 20% glucose (**left**) or YP + 10% glucose + 10% sucrose (**right**) medium for 2 d. Data represent mean values ± standard deviations from three independent experiments. Percentages in the graphs indicate how much emissions were affected by [*GAR^+^*]. Asterisks indicate that emissions significantly decreased by [*GAR^+^*] (*t* test, *p* < 0.05). YP: yeast extract–peptone medium; Glc: glucose; Suc: sucrose.

**Figure 2 ijms-25-00304-f002:**
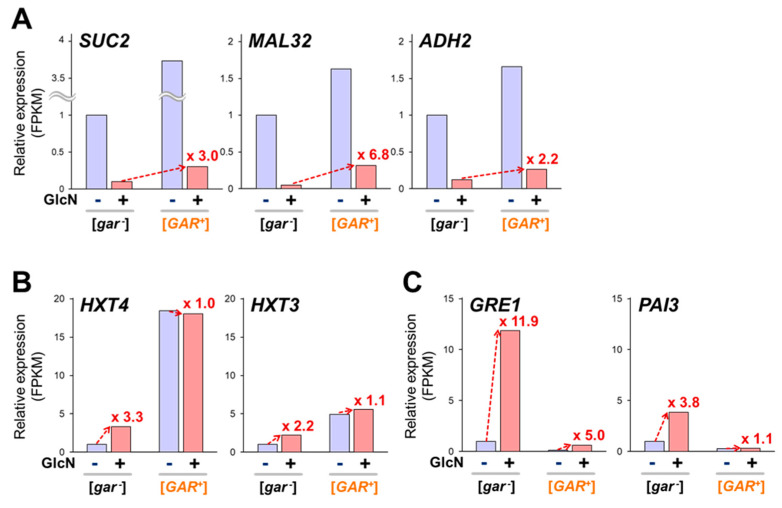
[*GAR^+^*] impairs transcriptomic responses to a nonmetabolizable glucose analog. Bar graphs represent the relative FPKM values for each gene normalized by the data of the [*gar^−^*] strain without addition of glucosamine (GlcN). Blue graphs show data without GlcN addition and red graphs show data with GlcN addition. Values in the graphs indicate fold changes in comparison to the data of each strain without addition of GlcN. Representative data of glucose-repressed genes (**A**), glucose-inducible hexose transporter genes (**B**), and glucose-inducible Hog1p target genes (**C**) are shown. FPKM: fragments per kilobase of exon per million fragments mapped.

**Figure 3 ijms-25-00304-f003:**
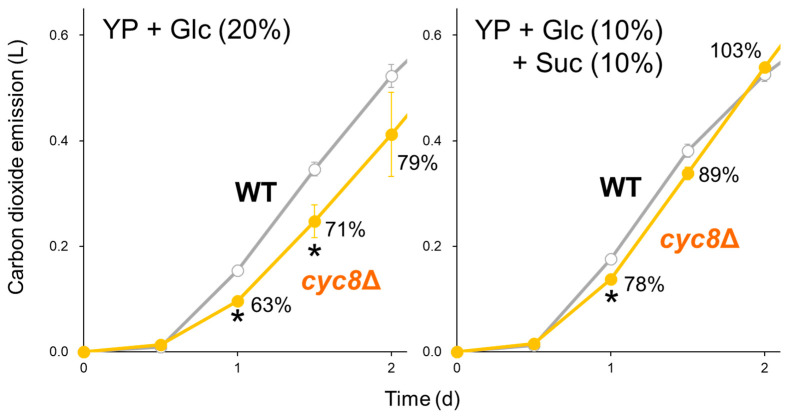
Deletion of the *CYC8* gene retards alcoholic fermentation when glucose is the sole carbon source. Carbon dioxide emissions of the wild-type (WT; gray) and *cyc8*Δ (orange) strains in the BY4741 background were monitored in YP + 20% glucose (**left**) or YP + 10% glucose + 10% sucrose (**right**) medium for 2 d. Data represent mean values ± standard deviations from three independent experiments. The percentages in the graph indicate how much emissions were affected by *cyc8*Δ. The asterisks indicate that emissions significantly decreased by *cyc8*Δ (*t* test, *p* < 0.05).

**Figure 4 ijms-25-00304-f004:**
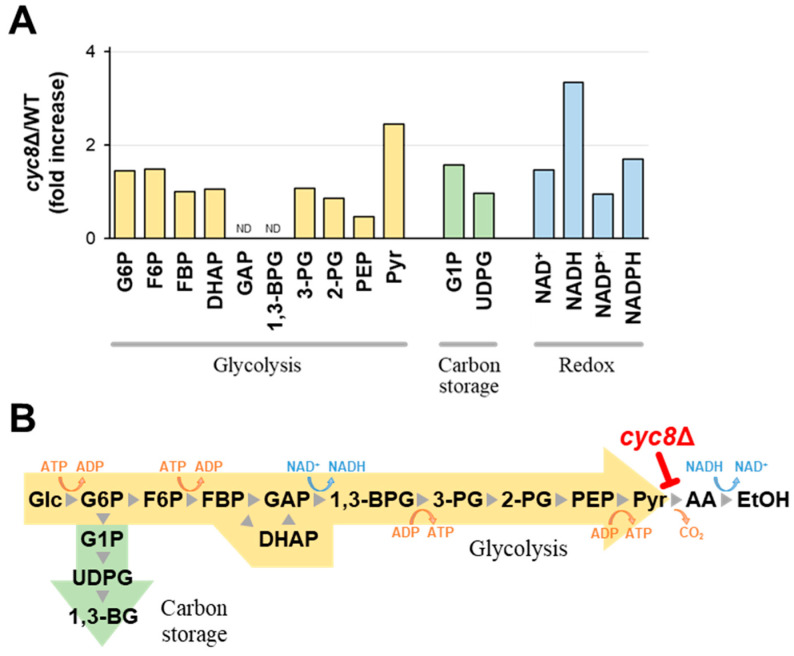
Deletion of the *CYC8* gene inactivates pyruvate decarboxylase. Bar graphs in (**A**) represent fold increases in each metabolite level per cell compared to that in WT. G6P; glucose 6-phosphate. F6P; fructose 6-phosphate. FBP; fructose 1,6-bisphosphate. DHAP; dihydroxyacetone phosphate. GAP; glyceraldehyde 3-phosphate. 1,3-BPG; 1,3-bisphosphoglycerate. 3-PG; 3-phosphoglycerate. 2-PG; 2-phosphoglycerate. PEP; phosphoenolpyruvate. Pyr; pyruvate. AA; acetaldehyde. EtOH; ethanol. G1P; glucose 1-phosphate; UDPG; UDP-glucose. ND; not detected. The schematic diagram in (**B**) indicates glucose metabolism in *S. cerevisiae* and the putative point of action of *cyc8*Δ. 1,3-BG; 1,3-β-glucan.

## Data Availability

The authors confirm that the data supporting the findings of this study are available within the article and its Appendix A.

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
