# Peer review of "Spontaneous Attenuation of Alcoholic Fermentation via the Dysfunction of Cyc8p in Saccharomyces cerevisiae"

_ijms, 2023, doi:10.3390/ijms25010304_

Round 1
Reviewer 1 Report (Previous Reviewer 1)
Comments and Suggestions for Authors
I have no issue with the paper as currently written. Although maybe the authors want to double check some of the grammar used in the edited sections.
In the response to my first review the authors show that the [GAR+] phenotype in their strains is not due to a chromosomal mutation. The authors don’t claim that [GAR+] is a prion, instead referring to is as “a non-chromosomal element such as a prion” or “prion-like [GAR+] factor”. I bring this up because the authors reply does not show that the element they are working with is indeed a prion. The claim of Brown and Lindquist that [GAR+] is a prion has not been substantiated by follow-up reports. The [GAR+] element might not be an infectious protein but rather some kind of metabolic programed loop. For a protein to be a prion there has to be infectivity. After all a prion is defined as an infectious protein. The tetrad data and the cytoduction data shown by Brown and Lindquist, that should proof that [GAR+] is a prion, are questionable. If [GAR+] was indeed a bona fide prion the presented work would be much more interesting.
Comments on the Quality of English LanguageMaybe the authors want to double check some of the grammar used in the edited sections.
Author Response
Thank you for your preferable review for our manuscript. We are aware that we must clearly prove that the spontaneous [GAR+] strains used in this study are indeed caused by a prion, and how it is generated from the viewpoint of molecular structures. Since we were able to clarify the possibility that Cyc8p is involved in this phenomenon in this paper, further analysis will be conducted focusing on the molecular state of the Cyc8p protein. We appreciate the reviewer's insightful comments, which will be helpful in our future research.
Reviewer 2 Report (Previous Reviewer 2)
Comments and Suggestions for Authors
The study of [GAR+] is an interesting topic but still needs to be answered. It was presumably believed to be a non-Mendelian, cytoplasmic inheritance initially but suggested to be the protein-based epigenetic element of inheritance or prion-based inheritance later. The manuscript described the RNA-Seq analysis from the X2180 parental [gar-] strain and its [GAR+] variant. The changes in gene expression patterns were recorded and analyzed in [GAR+] cells. The idea of Cyc8p involved in the formation of [GAR+] phenotype was postulated due to the Cyc8p–Tup1p complex was known to cooperatively interact with different proteins and regulate various physiological behaviors. The retardation of alcohol fermentation of the CYC8 mutant, TUP1 mutant, and the BY4741 wild type were also analyzed. The RNA-Seq data and intracellular metabolite profiles were solid, but many assumptions without experimental evidence existed. It is suggested that the following modifications be made before recommendation to the International Journal of Molecular Sciences.
1. The descriptions, in the abstract and in the discussion, that the stochastic mutation of Cyc8p may lead to the attenuation of alcoholic fermentation were inappropriate. The evidence was shown only based on a deletion CYC8 mutant in this study. The words “stochastic” or “stochastically” have to be carefully used before the data from the stochastic mutation of Cyc8p is available.
2. The descriptions that the function of the Cyc8p–Tup1p complex was impaired, altered, and involved in the [GAR+] phenotype were inappropriate because no direct evidence was shown in this study. Only the retardation of alcohol fermentation from the CYC8 mutant and the TUP1 mutant was observed. A double mutant is required to fulfill the above description. Either Cyc8p or Tup1p has independent regulatory functions.
3. The description of the key regulator that generates the [GAR+] phenotype in the last paragraph of the introduction section was inappropriate because there’s insufficient evidence to imply the master roles of Cyc8p or Tup1p or Cyc8p–Tup1p complex.
4. Different yeast strains were used in this study, which may lead to misinterpretations because they may have different genetic backgrounds. They have to be confirmed carefully. The laboratory yeast strain X2180 and a representative sake strain K701 were used to study attenuated alcoholic fermentation. The CYC8 mutant, TUP1 mutant, and the BY4741 wild-type were also used to study attenuated alcoholic fermentation, and the CYC8 mutant and BY4741 were further used to analyze the intracellular metabolite profiles. The amino acid sequences of CYC8 from kimoto-type sake strain Km67, a laboratory yeast strain S288C, and a sake strain K7 were aligned in Supplementary Figure 5 to show the deleted glutamine-rich region of CYC8 from strain Km67 to imply the roles of CYC8 were responsible for [GAR+] phenotype and prion formation. It is inappropriate unless the CYC8 amino acid sequences from [GAR+] and [GAR-] phenotypes of different laboratory strains or sake strains were obtained.
5. The source and the phenotype of the TUP1 mutant were not described in section 4.1.
6. It’s not precisely to describe that [GAR+] phenotype strain easily lost the phenotype when grown in the standard rich medium (page 3, lines 82-86). The culture time and the type of standard rich medium could be described.
7. A similar question as above. The X2180 parental [GAR-] strain and its [GAR+] variant were precultured in YPD medium overnight at 30°C and inoculated into 80 mL of YPGly medium for RNA-Seq analysis in section 4.3. Will an overnight culture, usually 12 hours, easily lead to the X2180 [GAR+] phenotype strain losing the phenotype in the YPD medium?
8. It’s not precisely to describe that the CUR1 and BTN2 genes, which encode anti-prion proteins, were highly expressed in the [GAR+] strain. The definition of high expression should be described. In addition, the BTN2 gene was not on the list of Supplementary Table 5. Was the BTN2 a highly expressed gene in the [GAR+] strain? Furthermore, the description of this idea is consistent with the increased expression of several chaperones in the [GAR+] strain, which could be further described by the types of chaperones and cited references.
9. The CYC8 was one of the genes with increased induction ratios in [GAR+] cells (> 2) in Supplementary Table 1. The expression pattern of the CYC8 gene can be further discussed as an example.
Author Response
Thank you for showing understanding of our study. We thoroughly modified the manuscript as below according to the reviewer’s insightful comments.
Comment 1. The descriptions, in the abstract and in the discussion, that the stochastic mutation of Cyc8p may lead to the attenuation of alcoholic fermentation were inappropriate. The evidence was shown only based on a deletion CYC8 mutant in this study. The words “stochastic” or “stochastically” have to be carefully used before the data from the stochastic mutation of Cyc8p is available.
Response: As the reviewer pointed out, we have not provided any evidence of stochastic inactivation of Cyc8p. We modified the expressions in the abstract (P.1 L.18) and in the discussion (P.6 L.214), according to the reviewer's comment. The word "stochastic" at P.6 L.206 was not removed because this sentence is a general description of the characteristics of alcoholic fermentation.
Comment 2. The descriptions that the function of the Cyc8p–Tup1p complex was impaired, altered, and involved in the [GAR+] phenotype were inappropriate because no direct evidence was shown in this study. Only the retardation of alcohol fermentation from the CYC8 mutant and the TUP1 mutant was observed. A double mutant is required to fulfill the above description. Either Cyc8p or Tup1p has independent regulatory functions.
Response: As the reviewer pointed out, we have not provided any direct evidence of the involvement of the Cyc8p–Tup1p complex in the [GAR+] phenotype. We modified the expressions in the discussion (P.7 L.234 and 260). However, we still assume that if Cyc8p is inactivated in the [GAR+] strain, the Cyc8p–Tup1p complex can be also defective. (Of course we understand the possibility that only the Cyc8p-specific functions were impaired in the [GAR+] strain, though.) Also, our RNA-seq data suggested that the transcriptomic characteristics of the [GAR+] strain can be possibly attributed to the impaired functions of the Cyc8p–Tup1p complex. Thus, the sentence in the discussion "Our findings suggest that [GAR+] is closely associated with loss of function of the Cyc8p–Tup1p complex." was modified into "Our RNA-seq data suggest that [GAR+] is closely associated with loss of function of the Cyc8p–Tup1p complex." (P.6 L.210).
Comment 3. The description of the key regulator that generates the [GAR+] phenotype in the last paragraph of the introduction section was inappropriate because there’s insufficient evidence to imply the master roles of Cyc8p or Tup1p or Cyc8p–Tup1p complex.
Response: We modified the expression in the introduction (P.2 L.64), according to the reviewer's comment.
Comment 4. Different yeast strains were used in this study, which may lead to misinterpretations because they may have different genetic backgrounds. They have to be confirmed carefully. The laboratory yeast strain X2180 and a representative sake strain K701 were used to study attenuated alcoholic fermentation. The CYC8 mutant, TUP1 mutant, and the BY4741 wild-type were also used to study attenuated alcoholic fermentation, and the CYC8 mutant and BY4741 were further used to analyze the intracellular metabolite profiles. The amino acid sequences of CYC8 from kimoto-type sake strain Km67, a laboratory yeast strain S288C, and a sake strain K7 were aligned in Supplementary Figure 5 to show the deleted glutamine-rich region of CYC8 from strain Km67 to imply the roles of CYC8 were responsible for [GAR+] phenotype and prion formation. It is inappropriate unless the CYC8 amino acid sequences from [GAR+] and [GAR-] phenotypes of different laboratory strains or sake strains were obtained.
Response: Thank you this suggestion. You have raised an important point here. However, we believe that the [GAR+] phenotype is not caused by a chromosomal mutation based on the previous papers (Brown and Lindquist, Genes Dev. (2009)). As far as we tested, the spontaneous [GAR+] strains do not contain mutations in the CYC8 gene in comparison to the parental [gar-] strains. Furthermore, BY4741 and X2180 are isogenic strains to S288C; BY4741 is part of a set of deletion strains derived from S288C in which commonly used selectable marker genes were deleted (Winston et al., Yeast (1995)). X2180 was generated via diploidization of S288C (Mortimer and Johnston, Genetics (1986)). Variation among these strains is miniscule. We believe that it is meaningful to compare amino acid sequences of the important regulators among different genetic backgrounds. For example, it is quite common to make comparisons between races with different genetic backgrounds to explore the causes of human disease. We confirmed that the deletion mutation in the amino-terminal glutamine-rich region of the CYC8 gene product in the Km67 strain (Supplementary Figure 5) is unique not only among X2180 and sake strains but also among the all known S. cerevisiae genomes shown in the Saccharomyces Genome Database (SGD; https://www.yeastgenome.org/).
Comment 5. The source and the phenotype of the TUP1 mutant were not described in section 4.1.
Response: We added information on the TUP1-deleted mutant in section 4.1 (P.7 L.275-277), according to the reviewer's comment.
Comment 6. It’s not precisely to describe that [GAR+] phenotype strain easily lost the phenotype when grown in the standard rich medium (page 3, lines 82-86). The culture time and the type of standard rich medium could be described.
Response: We added detailed information on the [GAR+] phenotypic reversion in section 2.1 (P.3 L.84-85), according to the reviewer's comment.
Comment 7. A similar question as above. The X2180 parental [GAR-] strain and its [GAR+] variant were precultured in YPD medium overnight at 30°C and inoculated into 80 mL of YPGly medium for RNA-Seq analysis in section 4.3. Will an overnight culture, usually 12 hours, easily lead to the X2180 [GAR+] phenotype strain losing the phenotype in the YPD medium?
Response: Thank you for the important points you raise. As we mentioned in section 2.1, the [GAR+] phenotype can be disappeared under non-selective conditions (i.e. medium without glucosamine). However, the RNA-seq analysis was performed before we realized this issue, so we precultured the strains in YPD medium. At that time, we decided the preculture medium based on Jarosz et al. (2014), stating that "Once acquired, the trait was maintained even after passage on nonselective glucose media for hundreds of generations". After that, we found that the [GAR+] phenotype can be reverted after the fermentation tests in YPD20 medium (5 days of duration) or sake fermentation (a few weeks of duration; we submitted an unpublished data in the previous "comments and responses"). Therefore, now we recognize that the [GAR+] strain may lose the phenotype during the long-term culture under fermentative conditions. We modified the expression in section 2.1 (P.3 L.84-85).
Comment 8. It’s not precisely to describe that the CUR1 and BTN2 genes, which encode anti-prion proteins, were highly expressed in the [GAR+] strain. The definition of high expression should be described. In addition, the BTN2 gene was not on the list of Supplementary Table 5. Was the BTN2 a highly expressed gene in the [GAR+] strain? Furthermore, the description of this idea is consistent with the increased expression of several chaperones in the [GAR+] strain, which could be further described by the types of chaperones and cited references.
Response: Although this part has been modified based on the comment 3 from the Reviewer #1 upon the first submission, we should have more carefully checked the data and description for the revised manuscript. As the reviewer pointed out, BTN2 did not show similar tendency to CUR1. The expression profiles of chaperone genes could not be simply attributed to the increased expression of CUR1. According to the reviewer's comment, we deleted this imprecise description. We truly appreciate the reviewer's important comment.
Comment 9. The CYC8 was one of the genes with increased induction ratios in [GAR+] cells (> 2) in Supplementary Table 1. The expression pattern of the CYC8 gene can be further discussed as an example.
Response: We added explanation on the expression pattern of the CYC8 gene in section 2.2 (P.4 L.141-145), according to the reviewer's comment.
Round 2
Reviewer 2 Report (Previous Reviewer 2)
Comments and Suggestions for Authors
The [GAR+] phenotype is interesting and needs comprehensive investigations to solve not only the academic research foundations but also the industrial production problems. It was presumably believed to be a non-Mendelian, cytoplasmic inheritance initially but suggested to be the protein-based epigenetic element of inheritance or prion-based inheritance later. The manuscript described the RNA-Seq analysis from the X2180 parental [gar-] strain and its [GAR+] variant. The changes in gene expression patterns were recorded and analyzed in [GAR+] cells. The idea of Cyc8p involved in the formation of [GAR+] phenotype was postulated due to the Cyc8p–Tup1p complex was known to cooperatively interact with different proteins and regulate various physiological behaviors. The retardation of alcohol fermentation of the CYC8 mutant, TUP1 mutant, and the BY4741 wild type were also analyzed. The RNA-Seq data and intracellular metabolite profiles were solid though still needed other experimental evidence. The revised manuscript was improved and corrected according to the suggestions and is recommended for the International Journal of Molecular Sciences. Below are the suggestions for further improvement.
1. The authors agreed with the comments that different yeast strains were used in this study, which may lead to misinterpretations because they may have different genetic backgrounds, and it is inappropriate unless the CYC8 amino acid sequences from [GAR+] and [GAR-] phenotypes of different laboratory strains or sake strains were obtained. It is suggested that the authors’ responses, quoted as shown below, to the text be added to clarify the issues.
(1) “BY4741 and X2180 are isogenic strains to S288C; BY4741 is part of a set of deletion strains derived from S288C in which commonly used selectable marker genes were deleted (Winston et al., Yeast (1995)). X2180 was generated via diploidization of S288C (Mortimer and Johnston, Genetics (1986)).” The references have to be cited according to the format of the journal.
In addition, the authors described that “As far as we tested, the spontaneous [GAR+] strains do not contain mutations in the CYC8 gene in comparison to the parental [GAR-] strains.” How did the authors test if whole-genome sequencing is not performed to rule out any mutations? It could be described in the text.
(2) “We confirmed that the deletion mutation in the amino-terminal glutamine-rich region of the CYC8 gene product in the Km67 strain (Supplementary Figure 5) is unique not only among X2180 and sake strains but also among all known S. cerevisiae genomes shown in the Saccharomyces Genome Database (SGD; https://www.yeastgenome.org/).”
In addition, it is suggested to show the sequence alignment of CYC8 proteins from different yeast strains, at least those related to the current study, to show the common deletion mutation in the amino-terminal glutamine-rich region of CYC8 proteins as supplementary materials.
2. The source and the phenotype of the TUP1 mutant were described in section 4.1. However, the strain names or the accession numbers of two mutants should be also provided. For example, Y07161 is for the CYC8 mutant, and Y07198 is for the TUP1 mutant.
Author Response
Thank you for giving me the opportunity to submit a revised draft of our manuscript titled "Spontaneous attenuation of alcoholic fermentation via the dysfunction of Cyc8p in Saccharomyces cerevisiae". We appreciate the time and effort that you and the reviewers have dedicated to providing your valuable feedback on our manuscript. We are grateful to the reviewers for their insightful comments on our paper. We have been able to incorporate changes to reflect most of the suggestions provided by the reviewers. We have highlighted the changes within the manuscript. Here is a point-by-point response to the reviewer's comments.
Comment 1. The authors agreed with the comments that different yeast strains were used in this study, which may lead to misinterpretations because they may have different genetic backgrounds, and it is inappropriate unless the CYC8 amino acid sequences from [GAR+] and [GAR-] phenotypes of different laboratory strains or sake strains were obtained. It is suggested that the authors’ responses, quoted as shown below, to the text be added to clarify the issues.
(1) “BY4741 and X2180 are isogenic strains to S288C; BY4741 is part of a set of deletion strains derived from S288C in which commonly used selectable marker genes were deleted (Winston et al., Yeast (1995)). X2180 was generated via diploidization of S288C (Mortimer and Johnston, Genetics (1986)).” The references have to be cited according to the format of the journal.
In addition, the authors described that “As far as we tested, the spontaneous [GAR+] strains do not contain mutations in the CYC8 gene in comparison to the parental [GAR-] strains.” How did the authors test if whole-genome sequencing is not performed to rule out any mutations? It could be described in the text.
Response: According to the comment, we added sentences in section 4.1 (page 8, lines 286-288). The sequences of CYC8 of the spontaneous [GAR+] strains were analyzed by Sanger sequencing. We mentioned this point in Discussion (page 7, lines 236-238).
(2) “We confirmed that the deletion mutation in the amino-terminal glutamine-rich region of the CYC8 gene product in the Km67 strain (Supplementary Figure 5) is unique not only among X2180 and sake strains but also among all known S. cerevisiae genomes shown in the Saccharomyces Genome Database (SGD; https://www.yeastgenome.org/).”
In addition, it is suggested to show the sequence alignment of CYC8 proteins from different yeast strains, at least those related to the current study, to show the common deletion mutation in the amino-terminal glutamine-rich region of CYC8 proteins as supplementary materials.
Response: According to the comment, we added a sentence in Discussion (page 6, line 232-page 7, line 236). We also added amino acid sequences of CYC8 gene products from BY4741 and X2180 in Supplementary Figure 5.
Comment 2. The source and the phenotype of the TUP1 mutant were described in section 4.1. However, the strain names or the accession numbers of two mutants should be also provided. For example, Y07161 is for the CYC8 mutant, and Y07198 is for the TUP1 mutant.
Response: According to the reviewer's comment, we revised section 4.1 (see page 7, lines 283-284).
This manuscript is a resubmission of an earlier submission. The following is a list of the peer review reports and author responses from that submission.
Round 1
Reviewer 1 Report
Comments and Suggestions for Authors
In this very interesting paper, the authors describe a potential link between the [GAR+] and [OCT+] elements of S. cerevisiae. A link between these two elements was already speculated in an earlier paper from this group (DOI: 10.1093/femsyr/foz061).
The molecular basis of [GAR+] is unknown although Brown and Lindquist (Genes Dev. 2009 23: 2320-2332, doi:10.1101/gad.1839109) suggested it to be a complex of Std1p and Pma1p. [OCT+] is an infectious protein formed by Cyc8 (DOI: 10.1038/ncb1843). Cyc8, in a complex with Tup1p, regulates the transcription of many genes. The authors identify targets of Cyc8-Tup1 that are also affected by the presence of [GAR+].
The data in this paper seems solid and I don’t have much to add. However, I am a bit concerned about the claim that the cells contain [GAR+]. Non-mendelian inheritance of the trait providing resistance to glucosamine when cells are grown on glycerol medium was first reported by Ball et al in 1976.
GLUCOSAMINE RESISTANCE IN YEAST. I. A PRELIMINARY GENETIC ANALYSIS
A J S Ball, D K Wong, J J Elliott
Genetics, Volume 84, Issue 2, 29 October 1976, Pages 311–317, https://doi.org/10.1093/genetics/84.2.311
Ball et al also identified two genetic loci that gave resistance to glucosamine on glycerol medium. Brown and Lindquist did not report finding any chromosomal mutants in their studies of [GAR+]. Unlike the other studies, Ball et al used UV mutagenesis in the experiment that identified a non-Mendelian element that provided resistance to glucosamine on glycerol medium. Nevertheless, their data indicate that chromosomal loci that can cause this phenotype do exist. Brown and Lindquist don’t mention how many isolates they analyzed. Brown and Lindquist also don’t mention how many tetrads of each isolate were analyzed when they claimed 4:0 non-Mendelian segregation of [GAR+]. The weak semi-dominance that Brown and Lindquist report is more indicative of a chromosomal mutation than of a prion.
The presence of a prion often gives a similar phenotype as mutations in the chromosomal gene that encodes the prion protein. However, when in such cases the prion more negatively affects the cells than the mutation of the chromosomal gene, it is possible that prolonged selection for the prion results in mutant containing cells starting to dominate the culture. Care should be taken to avoid this. In line with this the fact that the authors observed glucosamine resistant colonies appearing well above the general mutation rate is thus not a guarantee that the phenotype is due to a non-chromosomal element.
In the 2018 paper from the Takagi group, in which the isolation of the [GAR+] used in this paper is described, no attempt was made to verify that this prion was indeed isolated. As such it is plausible that the authors work with chromosomal mutants. If this would be the case their findings are still interesting. However, for the paper to be published as currently written, the authors should provide some proof that they are indeed working with a non-chromosomal element.
Other comments:
On line160 the claim is made that cells containing a CYC8 deletion grew to comparable levels as the WT. This data should be shown in the supplemental.
Line 193. The claim is made that increased expression of CUR1, which encodes an anti-prion protein, may support prion accumulation. This claim is a bit strange. How would increased expression of an anti-prion protein support prion accumulation? The reverse would be expected. In addition to CUR1 expression of BTN2 is also enhanced in the presence of glucosamine. As both Cur1p and Btn2p are degraded by the proteasome it seems likely that glucosamine causes a proteostasis stress. This idea fits with the increased expression of several chaperones.
Reviewer 2 Report
Comments and Suggestions for Authors
The manuscript described the elucidation of attenuated alcoholic fermentation via the impaired Cyc8p–Tup1p complex in Saccharomyces cerevisiae based on the data of RNA-Seq analysis from the X2180 parental [gar-] strain and its [GAR+] variant. The cells may spontaneously induce [GAR+] to generate non-genetic heterogeneity. The intracellular metabolite profiles of the CYC8-deletion mutant and its BY4741 wild type were also analyzed. Though the RNA-Seq data and intracellular metabolite profiles were reasonable, the study was presumptively questioned because of the lack of essential information. It is not suitable for publication in the International Journal of Molecular Sciences in its current form. Below are the major reasons.
1. Though Cyc8p–Tup1p complex was known to participate in regulating related gene expression under various glucose conditions, Cyc8p and Tup1p were also demonstrated to act independently, for example, the results revealed in a recent publication [25] (Lee et al., 2023; https://doi.org/10.1371/journal.pgen.1010876). The data showed from a CYC8-deletion mutant was not enough to prove that the attenuation of alcoholic fermentation was regulated by the Cyc8p–Tup1p complex. It needs to add evidence from a TUP1-deletion mutant and a CYC8 and TUP1 double mutant to demonstrate the postulation.
2. The laboratory yeast strain X2180 and a representative sake strain K701 were used to study attenuated alcoholic fermentation when glucose is the sole carbon source. However, the CYC8-deletion mutant and its BY4741 wild-type were used to analyze the intracellular metabolite profiles. Moreover, the amino acid sequences of CYC8 from kimoto-type sake strain Km67, a laboratory yeast strain S288C, and a sake strain K7 were aligned (Supplementary Figure 3). It is inappropriate that different laboratory strains and sake strains may have various genetic backgrounds. They may have different CYC8 sequences, even if they are laboratory strains or sake strains.
3. Section 2.3. It needs evidence to show the speculated conclusion that Cyc8p–Tup1p inactivates pyruvate decarboxylase for the decreased emission of carbon dioxide and acetaldehyde production.
Reviewer 3 Report
Comments and Suggestions for Authors
The research on the ability of yeast cells to assimilate various carbon sources, as well the effect when glucose represses the transcription of genes involved in the utilization of alternative carbon sources is important from the application point of view. The presented results are interesting and valuable.
The significance of the [GAR+ ] phenotype is good discussed
Line 253, detailed information about growth medium needed (producer, city, country)
The same requirements to all media used in the experiment
The literature could be updated, only five issues are from the last three years.